# Generating Transferable Adversarial Simulation Scenarios for Self-Driving via Neural Rendering

**Yasasa Abeysirigoonawardena**[†1], **Kevin Xie**[1,2], **Chuhan Chen**[3], **Salar Hosseini**[1],
**Ruiting Chen**[1], **Ruiqi Wang**[4], **Florian Shkurti**[1,2,5]
[1]University of Toronto, [2]Vector Institute, [3]Carnegie Mellon Universty,
[4]Stanford University, [5]UofT Robotics Institute

**Abstract:** Self-driving software pipelines include components that are learned from a significant number of training examples, yet it remains challenging to evaluate the overall system's safety and generalization performance. Together with scaling up the real-world deployment of autonomous vehicles, it is of critical importance to automatically find simulation scenarios where the driving policies will fail. We propose a method that efficiently generates adversarial simulation scenarios for autonomous driving by solving an optimal control problem that aims to maximally perturb the policy from its nominal trajectory. Given an image-based driving policy, we show that we can inject new objects in a neural rendering representation of the deployment scene, and optimize their texture in order to generate adversarial sensor inputs to the policy. We demonstrate that adversarial scenarios discovered purely in the neural renderer (surrogate scene) can often be successfully transferred to the deployment scene, without further optimization. We demonstrate this transfer occurs both in simulated and real environments, provided the learned surrogate scene is sufficiently close to the deployment scene.

## 1 Introduction

Safety certification of a self-driving stack would require driving hundreds of millions of miles on real roads, according to [1], to be able to estimate miles per intervention with statistical significance. This could correspond to decades of driving and data collection. Procedural generation of driving simulation scenarios has emerged as a complementary approach for designing unseen test environments for autonomous vehicles in a cost-effective way. Currently, generation of simulation scenarios requires significant human involvement, for example to specify the number of cars and pedestrians in the scene, their initial locations and approximate trajectories [2], as well as selection of assets to be added to the simulator. In addition to being challenging to scale, having a human in the loop can result in missing critical testing configurations.

In this paper, we cast adversarial scenario generation as a high-dimensional optimal control problem. Given a known image-based driving policy that we want to attack, as well as the dynamics of the autonomous vehicle, we aim to optimize a photorealistic simulation environment such that it produces sensor observations that are 3D-viewpoint-consistent, but adversarial with respect to the policy, causing it to deviate from its nominal trajectory. The objective of the optimal control problem is to maximize this deviation through plausible perturbations of objects in the photorealistic environment.

Our optimal control formulation requires differentiation through the sensor model in order to compute the derivative of the sensor output with respect to the underlying state perturbation. However, most existing photorealistic simulators for autonomous vehicles are not differentiable; they can only be treated as black boxes that allow forward evaluation, but not backpropagation. Instead of using an off-the-shelf photorealistic simulator and adding assets to match the scene, we train an editable

---

[†]Correspondence to `yasasa@cs.toronto.edu`

7th Conference on Robot Learning (CoRL 2023), Atlanta, USA.

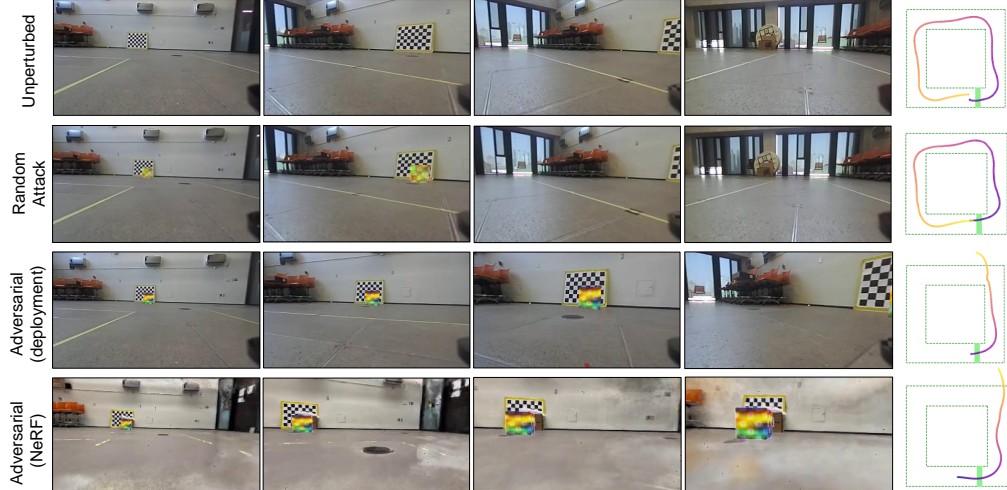

Figure 1: First-person-view (FPV) of our adversarial attack transfer to an RC car with overhead trajectory view on the right. Row 1: Unperturbed policy execution; Row 2: Random search texture attack; Row 3: Our adversarial attack directly transferred to the real deployment scene, without additional optimization; Row 4: Our adversarial attack discovered in the surrogate NeRF simulator.

neural rendering model that imitates the deployment scene, allowing us to insert new objects in the simulator and to optimize their texture through gradient-based optimization. This editable neural rendering model acts as a surrogate physics and rendering simulator, enabling us to differentiate through it in an efficient way in order to attack the driving policy's input sensor observations.

Unlike many existing types of adversarial attacks in the literature [3, 4, 5], our work aims to discover environment perturbations/attacks that satisfy the following properties: (a) **They are temporally-consistent**. The influence of the attack is not instantaneous, it is amortized through time via the optimal control loss function. (b) **They are transferable**. An attack discovered in the surrogate scene should ideally be adversarial in the actual deployment scene. (c) **They are object-centric**. The attack introduces and edits objects as opposed to unstructured high-frequency perturbations across all pixels. Specifically, we make the following contributions:

1. We formulate adversarial scenario generation across time as an optimal control problem that relies on a learned, surrogate NeRF simulator. The solution to this problem yields 3D-view-consistent, object-centric, adversarial attacks that often transfer to the deployment environment. We show how to solve this problem efficiently using implicit differentiation.
2. Differentiable rendering of our surrogate NeRF model enables gradient-based adversarial object insertion and scales to high dimensional parameterizations of multiple objects.
3. We show that our adversarial attacks discovered in the surrogate NeRF simulator can be realized in the real-world and retain their ability to disrupt the policy.

We experimentally validate our framework by reconstructing scenes using only pose-annotated images and generate adversarial object insertion attacks with multiple trajectories.

## 2   Related Work

**Adversarial scenarios for autonomous driving**. Perceptual adversarial attacks make modifications to prerecorded sensor data from real driving sessions to fool the perception system. Since this sensor data is fixed, they lack the ability to resimulate and typically only operate on the individual frame level. Previous works, [4, 6] attempt to attack a LiDAR object detection module by artificially inserting an adversarial mesh on top of car rooftops or objects in a prerecorded LiDAR sequence. They extend the scope of their attack further by incorporating textures to be able to attack image-based object detectors as well [3]. In both these works, the inserted object has a very low resolution

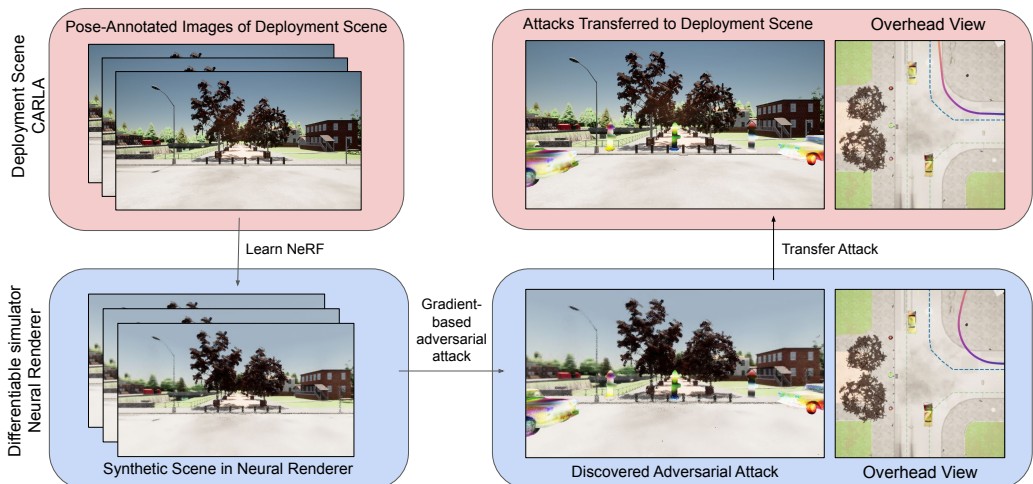

Figure 2: Our method can be summarized in the four steps shown. (a) In the top left, we obtain posed images from the deployment scene which can be a simulator or the real world. (b) In the bottom left, we reconstruct a surrogate scene by fitting a NeRF to the posed images as a differentiable simulator and observe only minor perceptual gap. (c) Having the surrogate scene, we can insert objects, which are also represented as NeRFs, and attack their color fields to generate textural attacks. (d) The discovered adversarial objects are introduced back into the deployment scene.

and nondescript geometry. Recent self-driving simulators, such as DriveGAN [7], GeoSim [8] and UniSim [9] address these issues, with the latter enabling manipulable sensor-based simulators based on prerecorded datasets. These works, however, have not dealt with discovering attacks.

Another prominent line of works produce dynamic state-level adversarial attacks. These generally target the control/planning system only by perturbing trajectories of other agents in the scene. Without considering the perception system, these methods use simplified traffic and state-based simulators that do not incorporate 3D rendering [10, 11, 12].

Closest to our work, a few methods have proposed to attack end-to-end policies by adding perturbations to existing self-driving simulators. In [13], the trajectories of other agents in a CARLA scene are modified to generate a collision event. Due to the non-differentiability of the simulator, a black-box Bayesian optimization is used. Gradient-based attacks on top of simulators have also been investigated. However, the requirement of differentiability has so far limited their scope to very simplified geometries that are composited post-hoc onto renderings from CARLA. In [5], flatly colored rectangles are composited on top of frames from the CARLA simulator and optimized to cause maximal deviation of an end-to-end image-based neural network steering controller. Similarly, work in [14] attempts to play a video sequence of adversarial images on a billboard in the scene using image composition. To our knowledge, no works in this setting have been able to demonstrate transfer of adversarial attacks to the real world, as these attacks rely on a pre-existing simulator that they augment. Compared to these, our attacks are entirely performed on a surrogate neural simulator that is reconstructed from only posed images captured from any deployment scene. Furthermore, our surrogate neural simulator allows for inserting arbitrary objects reconstructed from posed images.

The driving simulator VISTA [15] generates high fidelity sensor data using a large collection of real world data. In our case, we are able to train a NeRF using the data, allowing us to generalize to a wider range of novel views. [16] samples adversarial masking of existing LiDAR data using reinforcement learning. Work on perception error models [17] avoids using a simulator altogether and instead focuses on learning a surrogate policy that uses lower dimensional salient features, which are attacked. However, it would be very difficult to infer the real world perceptual disturbance that would cause the attack, so these attacks are very challenging to transfer to the real world.

**Robust adversarial examples.** Adversarial attacks for classification have commonly used minimal perturbations on the input images [18] that may not always transfer to the physical world or another

domain. To enhance robustness to domain transfer [19] proposes a class of adversarial transformations that optimize for adversarial attacks under a distribution of image transforms [20].

## 3  Background

### 3.1  Neural Rendering

Neural 3D representations, such as neural radiance fields (NeRF), have seen significant activity in recent years due to their ability to reconstruct real world objects and scenes to very high detail using only posed images. A survey of recent progress in neural rendering can be found in [21].

In [22], physics simulations of known objects are combined with their learned NeRF to create high fidelity dynamic environments for training deep reinforcement learning agents that can transfer to the real world. In our work, we use composition of NeRFs to insert and optimize adversarial objects. This is shown in Fig. 3, and the details are in Sections 4.1 and 4.2. We render the scene using the volume rendering equation:

$$I(x, \omega) = \int_0^T \sigma(x + t\omega) \exp\left(\int_0^t \sigma(x + \hat{t}\omega)\mathrm{d}\hat{t}\right) L(x + t\omega, -\omega)\mathrm{d}t \tag{1}$$

Where $I(x, \omega)$ is the intensity at a location $x$ given in world space in the direction $\omega$. $L$, and $\sigma$ are the learned color and density fields in NeRF. For the sake of performance, we choose to use grid-based volume representations. Structured grid NeRFs reduce computation cost by storing direct density and color variables [23] or latent features [24, 25] on explicit 3D grids. In essence, they trade extra memory utilization for large performance improvements. Instant Neural Graphics Primitives (iNGP) [26] uses multi-scale dense grids and sparse hash grids of features that are decoded to color and density by a MLP. We chose to use iNGP because it balances our performance and memory tradeoffs well.

## 4  Method

Our framework generates successful adversarial attacks of end-to-end image-based self-driving policies with only access to posed images from the deployment scene. An overview of the high-level steps in our framework is shown in Figure 2.

We now briefly describe the setting and our adversarial attack method. More details are included in Appendix B. Let $x_t$ denote the state of the car at time $t$, $x^*$ denote a reference trajectory to track and CTE the cross-track error. Starting from Eqn. 6, we set the cost function $C(x_t)$ of our problem as the car's proximity to the reference $x^*$:

$$C(x_t) = -\mathrm{CTE}(x_t, x^*) \tag{2}$$

In other words, we want to maximize deviation from the desired trajectory. We set the constraint function $G(x_t, x_{t+1}, \theta) = 0$ to be the following set of constraints:

$$u_t = \pi_\phi(o_t) \qquad (3) \qquad\qquad o_t = h_{\gamma,\theta}(x_t) \qquad (4) \qquad\qquad x_{t+1} = f_c(x_t, u_t) \qquad (5)$$

Where $\pi$ is the fixed driving policy*, $h$ is the neural rendering sensor model that outputs image observations $o_t$ given the state of the car. The renderer depends on $\theta$, the parameters of adversarial NeRF objects and $\gamma$, the fixed rendering parameters of the background scene NeRF. Finally, $f_c$ denotes the dynamics of the ego vehicle that must be considered, since we want to find adversarial trajectories that are consistent across multiple frames.

### 4.1  Differentiable Renderer

Traditional simulators like CARLA do not admit computation of gradients. Thus, prior works rely on artificially compositing simplistic textured geometries on top of rendered images from CARLA and

---

*We train our own policy and provide details in Appendix C.2.

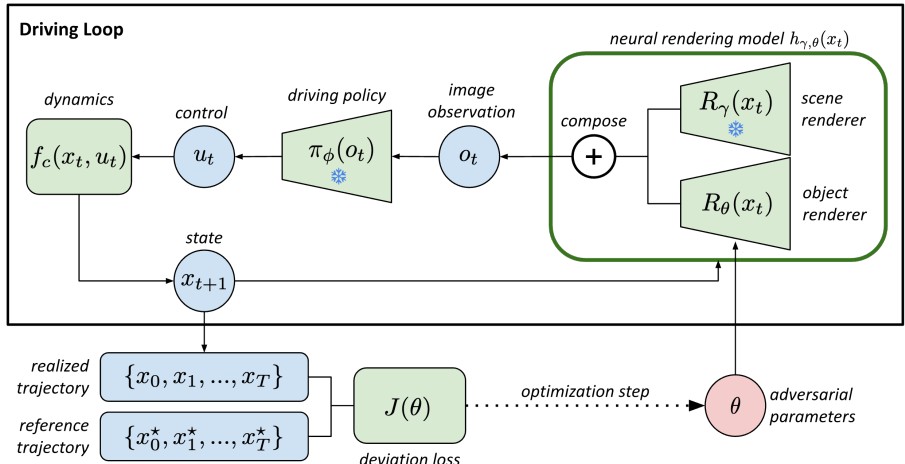

Figure 3: A computation diagram of our algorithm for generating adversarial attacks. The inner driving loop consists of three components: the neural rendering model, the differentiable driving policy, and the differentiable kinematic car model. We inject the adversarial perturbation to the surrogate scene by composing the outputs of one or more neural object renderers (the single object case is shown above for simplicity) with the output of the neural scene renderer. The parameters of the object renderer(s) are optimized to maximize the deviation of the realized trajectory from the reference trajectory, while keeping the parameters of the driving policy and scene renderer frozen.

obtaining gradients with respect to the composited alteration [14]. We use NeRFs to learn surrogate models of the scene and sensor model instead. This surrogate model not only gives us an automated method to reconstruct scenes from pose-annotated images, but also provides efficient gradient computation giving us a differentiable form for the sensor $h$. For the purposes of optimization, we found traditional NeRF representations to be intractable in terms of compute and memory requirements (during gradient computation). Thus, we opt to use the multi-resolution hash grid representation, Instant-NGP [26].

Note that, similar to existing work, we detach the gradients of the image observation with respect to the camera coordinates (which are attached to the ego vehicle) [27]. We include more details regarding this in Appendix B.3.

## 4.2 Adversarial Object Insertion

We use insertion and texturing of multiple objects as our adversarial perturbations to the background scene. To do this, we first reconstruct regular objects, such as cars, as individual NeRFs from pose-annotated images. For our object NeRFs we simply store color values directly on the voxel grids of Instant-NGP, which are tri-linearly interpolated within each voxel. By choosing these color voxel grids as our adversarial parameters $\theta$, we can perform independent adversarial texture attacks over multiple objects.

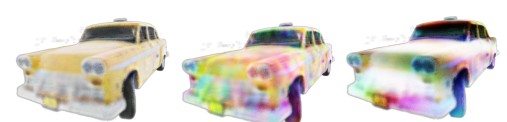

The object NeRFs can be easily composed with our background scene NeRF. This is done via alpha compositing, which leverages opacity and depth values that can be easily computed.

Figure 4: Base car on the left; random texture in the middle; adversarial texture on the right.

## 4.3 Gradient computation via implicit differentiation

We use implicit differentiation for gradient computation [28], also known as the adjoint method, which enables constant memory gradient computation with respect to trajectory length. In discrete time, the adjoint method amounts to propagating gradients through an implicit relationship $G$ for

problems of the form:

$$\min_\theta J(\theta) = \sum_{t=0}^{T} C(x_t) \text{ such that } G(x_{t-1}, x_t, \theta) = 0 \tag{6}$$

Explicitly, the method performs a forward simulation to compute the variables $x_t$ and then subsequently a backward pass to compute adjoint variables $\lambda_t$ by solving the equations:

$$\frac{\partial G(x_{t-1}, x_t)}{\partial x_t}^\top \lambda_t = -\frac{\partial C(x_t)}{\partial x_t}^\top - \frac{\partial G(x_t, x_{t+1})}{\partial x_t}^\top \lambda_{t+1} \tag{7}$$

with the boundary condition:

$$\frac{\partial G(x_{T-1}, x_T)}{\partial x_T}^\top \lambda_T = -\frac{\partial C(x_T)}{\partial x_T}^\top \tag{8}$$

Finally, the gradient of the loss can be calculated as:

$$\nabla_\theta J = \lambda_1^\top \frac{\partial G(x_0, x_1, \theta)}{\partial x_0} \frac{\partial x_0}{\partial \theta} + \sum_{t=1}^{T} \lambda_t^\top \frac{\partial G(x_{t-1}, x_t, \theta)}{\partial \theta} \tag{9}$$

Throughout both passes we do not need to store large intermediate variables and only need to accumulate the gradient at each step.

## 4.4 Gradient-based Adversarial Attack

Obtaining gradients for the problem in Eqn. (6) should be possible with an autodifferentiation framework such as PyTorch [29]. We find that naively computing the gradient via backpropagation results in memory issues as we scale up trajectory lengths due to all the intermediary compute variables used to compute the integral in Eqn. 1 being stored until the end of the trajectory. We achieve drastic memory savings by using the adjoint method [30] which only keeps track of the adjoint variables $\lambda$ along the trajectory. In our case, the adjoint variables are three-dimensional, allowing us to only use as much memory as it takes to compute a single jacobian vector product of the composition of models given by (5), (3), (4) in the optimization problem in Eqn. (6).

To summarize, the computation of our gradient-based adversarial attack proceeds as follows:
1. We rollout our policy in our surrogate simulator to compute the loss and the trajectory $x_{1:T}$ in Eqn. (6).
2. We perform a backward pass to compute adjoint variables for gradient computation.
3. Using the adjoint variables, we compute the gradient $\nabla_\theta J$ and update parameters $\theta$.

## 5 Experiments

To demonstrate the effectiveness of our framework, we aim to reconstruct a driving scenario from posed images, generate adversarial attacks and validate that those attacks transfer to the deployment scene. Through our experiments, we would like to answer the following key questions:
- (Q1) Can gradient based optimization find better adversarial examples than random search?
- (Q2) Are NeRF models suitable surrogate renderers for gradient based adversarial optimization?
- (Q3) Are adversarial attacks transferable from NeRF back to the deployment domain?

## 5.1 Experimental Details

**CARLA Deployment Scenes.** We first validate our method in simulation, treating CARLA as a proxy for a real deployment scene. We perform experiments on a 3-way intersection of the CARLA [31] simulator. For the 3-way intersection, we consider 3 different trajectories to be followed by the ego vehicle. For the object models, we train surrogate NeRF models for two sample objects, a fire hydrant and a small car using only posed images (any other object could be used). We

manually insert 2 small cars and 3 fire hydrants into the driving scene in an initial placement. Our adversarial attacks jointly optimize the NeRF color parameters and object rigid transforms.

**Real World Deployment Scenes.** Our real-world experiments are performed on an autonomous RC car driving around a square race track in an indoor room. It is difficult to manufacture adversarial attacked objects with complex shapes in the real world. Hence, for practicality, we insert a NeRF object representing a flat square texture pattern that can be projected by a display monitor in the real world and optimize its color parameters. In order to create a first version of the attack we choose to directly compose the adversarial texture on to the robot camera feed. We then move on to a more difficult task of physically realizing this attack, for this we opted to display the texture on a monitor to simplify lighting conditions. Additional details of our real world experimental setup are given in C.4.1

## 5.2 Evaluation Metrics

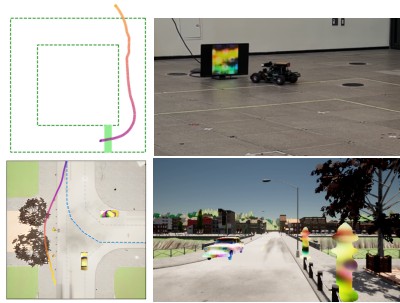

Figure 5: Selected overhead views and snapshots from adversarial deployment trajectories in the real world (top row: monitor displays adversarial texture discovered in NeRF), and in CARLA (bottom row: adversarial objects inserted in the simulator).

We measure the effectiveness of an attack with our adversarial objective, the cross track error of the vehicle. We use the road center as the reference and so even an unperturbed driving policy has some non-zero deviation which we report under "Unperturbed" in Table 1. To characterize the insensitivity of our method to random seeds, we run 5 separate attacks per scenario for both the gradient-based and random attacks with different random initializations of the adversarial parameters. We report the mean and standard deviation of our metric.

Our proposed method of attack is via gradient-based optimization using the method outlined in Section 4.4. The gradient-based attack uses 50 iterations of optimization using Adam, with a learning rate of 0.1. Due to the high dimensional parameterization, detailed in B.3.1, bayesian optimization becomes computationally intractable. Therefore, as a baseline for our method, we perform a random search parameter attack on the NeRF surrogate model that samples parameters from a Gaussian distribution with mean zero and a standard deviation of 5. We chose this standard deviation to match the distribution over parameters we found in our gradient attacks. We use the same number of function evaluations, selecting the best achieved attack among the 50 random samples for the CARLA experiment. For real-world experiments, we didn't find much variation between random attacks in the surrogate simulator, showing the difficulty of random search in high dimensional parameter spaces.

## 5.3 Experimental Results

Example gradient attack trajectories are shown in Figure 5. We include more visualization of results for deployments of adversarial attacks, both in CARLA simulation in the real world and preliminary results of retraining the CARLA policy using new data, in Appendix D. In Table 1 we compare the total cross track errors caused by our adversarial attack against the expert lane following controller.

We observe in all 3 CARLA scenarios (averaged over 5 seeds each) that our adversarial attacks using gradient optimization consistently produce significant deviation from the lane center. When transferring these attacks back into the deployment scene, we see that although the magnitude of the deviation is reduced, we still retain a significant increase over the unperturbed or random search setting. The difference is likely due to visual imperfections in our surrogate NeRF simulator compared to the deployment scene. The random search perturbations are far less effective, remaining near the baseline unperturbed trajectory for 2 out of the 3 cases.

| | CARLA Deployment | Surrogate Scene | | CARLA Deployment | |
|---|---|---|---|---|---|
| Scenario | Unperturbed | Random | Gradient | Random | Gradient |
| Straight | 1166 | $1132 \pm 7$ | $2347 \pm 49$ | $1193 \pm 19$ | $1702. \pm 160$ |
| Right | 1315 | $2084 \pm 10$ | $4105 \pm 847$ | $1476 \pm 12$ | $2101. \pm 75$ |
| Left | 1448 | $1460 \pm 8$ | $4125 \pm 124$ | $1158 \pm 163$ | $2240. \pm 574$ |
| | Physical Deployment | Surrogate Scene | | Physical Deployment | |
| Setup | Unperturbed | Random | Gradient | Random | Gradient |
| Green Screen | 48 | $34 \pm 4$ | $157 \pm 1$ | $46 \pm 3$ | $248 \pm 72$ |
| Monitor | | | | $47 \pm 3$ | $76 \pm 48$ |

Table 1: Comparison of the total cross-track error for all the scenario tested. Results are shown for the following cases: (1) no attack in the deployment scene (unperturbed), (2) an adversarial attack (random or gradient) in the surrogate NeRF scene, (3) an attack in the deployment scene. We separate results from the CARLA and physical deployments, we show that gradients in our surrogate simulator are useful for finding adversarial attacks and these attacks remain effective when transferred to the deployment environment.

For the real world experiment, we observed a similar result. Random attacks consistently fail to elicit deviation from the driving policy both in the surrogate and deployment scenes. Over 5 random seeds, not a single random attack was able to cause the vehicle to exit the track. Gradient attacks on the other hand are reliably able to find strong attacks with little variance in the surrogate scene. When transferring our attacks to the real world, we find the attacks to retain their strength in the green screen setup. The strength of the attack is relatively diminished when using the monitor to project the attack but is nonetheless consistently higher than the random attack and causes the vehicle to understeer and exit the track on occasion. We suspect this is due to the display properties of the monitor which can alter the appearance of the adversarial perturbation.

## 6 Limitations

Despite showing the ability to generate 3D-consistent adversarial scene perturbations, there are a few avenues for improvement. First, we assume that the vision-based driving policy is differentiable. Recent works have shown high potential for modular end-to-end learned policies that could leverage synergies between vision and planning, such as neural motion planning [32], transfuser [33] and many others [34]. We discuss three potential methods to handle non-differentiable policies in Appendix E.1. Second, while we do optimize for both adversarial textures and object poses, our experiments in section D.3 of the Appendix show that the latter produces significantly non-smooth loss landscapes that necessitated multi-start gradient optimization methods to handle local minima.

## 7 Conclusion

We presented a method for generating 3D-consistent object-based adversarial perturbations in autonomous driving scenarios. Unlike previous approaches that rely on making edits on top of fixed pre-recorded data or black-box simulators, we develop a differentiable simulator directly with a neural radiance field representation of geometry and texture of a scene that admits gradients through the rendering of camera and depth observations. Through alpha-compositing, we can introduce new objects also represented as neural radiance fields into the scene and optimize color perturbations of the objects. We validate our framework both in simulation and on a real-world RC car race track driving scenario showing successful sim-to-real transfer of discovered attacks. While our particular implementation is only a first step towards demonstrating NeRF based adversarial attack generation, we believe that our framework shows a promising new direction for automatic evaluation of autonomous vehicles. We expect our method to benefit greatly from continued improvements being made to neural rendering and their wider adoption for AV/robotic simulation.

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

# A Appendix: Additional Background

## A.1 Neural Rendering

**Differentiable rendering**. NeRFs represent scenes as emissive volumetric media [26]. Unlike surface rendering, volumetric rendering does not suffer from explicit hard discontinuities, which are difficult to handle for traditional surface rendering methods[21]. We exploit the differentiable volume rendering of NeRFs, to robustly compute efficient gradients for arbitrary geometries.

**Complex scene reconstruction**. Works such as BlockNeRF [35] and S-NeRF [36] show great potential for automatically capturing street-level scenes relevant to autonomous vehicle simulation. Unlike traditional simulators, these neural representations are directly trained on raw sensor captures, thereby obtaining high-fidelity visual reconstruction without laborious asset creation.

**Composition and editing**. Recent works have extended the static single scene setting of NeRF to composition of NeRFs, scene disentanglement, as well as editing and relighting. Specifically, [37] encodes scenes with latent codes from which new scenes can be generated. [38], [39] and [40] introduce compositional radiance fields to represent different objects and realize scene decomposition. [41] utilizes 2D segmentation information to perform 3D scene inpainting. [42] and [43] decompose color into different illumination components. [44] [45] learn priors from big datasets of images to disentangle existing scenes.

**Control and neural rendering models**. Neural rendering has seen utility in a few different optimization tasks for robotics such as pose estimation [46]. NeRFs have also seen direct application to trajectory optimization, including utilizing NeRF's density as approximate occupancy [47], collision, and friction constraints [48], effectively allowing NeRF to act as a differentiable physics simulator. On the other hand, [49] learns an additional latent dynamics of NeRF objects. In [22], physics simulations of known objects are combined with their learned NeRF to create high fidelity dynamic environments for training deep reinforcement learning agents, that can transfer to the real world. We use composition of NeRFs in our work to insert and optimize adversarial objects. This is shown in Fig. 3, and the details are in Sections 4.1 and 4.2.

# B Appendix: Method Details

## B.1 NeRF

**Volume Rendering** A neural radiance field consists of two fields, $\sigma_\phi(x), L_\psi(x, \omega)$ that encode the density $\sigma$ at every location $x$ and the outgoing radiance $L$ at that location in the direction $\omega$. In NeRFs, both of these functions are represented by parameterized differentiable functions, such as neural networks. Given a radiance field, we are able to march rays through an image plane and reconstruct a camera image from a given camera pose and intrinsic matrix using the rendering function reiterated here for clarity:

$$I(x, \omega) = \int_0^T \sigma(t) \exp\left( \int_0^t \sigma(\hat{t}) d\hat{t} \right) L(t, -\omega) dt \qquad (B.1)$$

Where $L(t, \cdot)$ and $\sigma(t)$ are shorthands for $L(t\omega + x, \cdot)$ and $\sigma(t\omega + x)$, and $I(x, \omega)$ is the intensity at a location $x$ given in world space in the direction $\omega$.

**Compositing** For our adversarial attacks to contain 3D semantics, it is crucial to insert the perturbation in a 3D aware manner. For this we utilize another feature of neural radiance fields, which is to output opacity values. Specifically, in Eqn. (1) we can extract the transmittance component, which acts as a measure of the pixel transparency $\alpha$:

$$\alpha(x, \omega) = \exp\left( \int_0^t \sigma(\hat{t}) d\hat{t} \right) \qquad (B.2)$$

Furthermore, we can replace the radiance term with distance in (1) to extract the expected termination depth of a ray $z$:

$$z(x, \omega) = \int_0^T t\sigma(t)\alpha(t)\mathrm{d}t \tag{B.3}$$

We consider the case of two radiance fields, the object radiance field $\sigma_o, L_o$ and the background radiance field $\sigma_s, L_s$. We use a transformation matrix to correspond ray coordinates between the scene and the object radiance field.

By applying equations (1), (B.2), (B.3) to a single ray that corresponds to both the base scene and the object radiance field, we obtain the values $c_o$, $\alpha_o$, $z_o$, $c_s$, $\alpha_s$, $z_s$ respectively, where $\alpha_*$ is the opacity and $z_*$ is the depth along the ray. We denote the foreground and background values at a pixel as

$$f = \underset{o,s}{\arg\min}(z_s, z_o) \tag{B.4}$$

$$b = \underset{o,s}{\arg\max}(z_s, z_o) \tag{B.5}$$

The final blended color is then given by:

$$c = \frac{\alpha_f c_f + (1 - \alpha_f)\alpha_b c_b}{\alpha_f + \alpha_b(1 - \alpha_f)} \tag{B.6}$$

In the case of multiple object NeRFs, we simply repeat the alpha blending for each object to composite them all into the same scene.

## B.2 Vehicle Dynamics

The dynamics in equation (5) can take multiple forms, for the CARLA experiments, we choose the simplest kinematic model of a car, a Dubin's vehicle:

$$\dot{x} = \begin{bmatrix} v\cos\theta \\ v\sin\theta \\ u \end{bmatrix} \tag{B.7}$$

For the purposes of the CARLA deployment environment, we find that it is sufficient to consider the kinematic model with fixed velocity, and only angular control. Thus, our imitation learning policy in Eqn. (3) only outputs steering commands. We note that our approach is applicable to any dynamics model, as long as it is differentiable.

For the real world experiments, we opted for a fixed velocity Ackerman steering model:

$$\dot{x} = \begin{bmatrix} v\cos\theta \\ v\sin\theta \\ \frac{v}{l}\tan(\theta) \end{bmatrix} \tag{B.8}$$

where $l$ is the robot wheelbase.

## B.3 Optimization Details

As described in Section 4.1, following prior work, we do not propagate gradients of camera parameters through the sensor model function. Specifically, we set,

$$o_t = h_{\gamma,\theta}(\text{stop\_gradient}(x_t)) \tag{B.9}$$

Thus gradients of the observation will only be taken with respect to the adversarial object parameters $\theta$ and not the state of the car. The gradient with respect to $x_t$ corresponds to exploiting higher order effects of how the observation would change if the car was looking in a slightly different direction due to previous steps of the attacks, and leads to a very non-smooth loss objective that is not useful for finding practical attacks.

For experiments in the real world, we found the attacks were sometimes very sensitive to the robot's pose. To alleviate this issue, we chose to optimize multiple randomly sampled initial poses simultaneously. The samples were normally distributed around the nominal car starting location, with a standard deviation of 0.1.

### B.3.1 Optimization parameters

In all our experiments, our optimization parameters $\theta$ correspond to values on the NGP voxel grid. Since we have removed the decoder, the grid values directly correspond to the color for a given position in the volume. Due to this, the parametrization even for small models can get quite large, in the order of a 5 million for the hydrant.

## C   Appendix: Experimental Details

### C.1   NeRF Models

When training the surrogate NeRF models of the background scene and objects, we use the default Instant-NGP hyperparameters and optimize over 50 epochs using the Adam optimizer.

The source 3D assets for our objects were obtained from the Objaverse dataset [50] and posed images produced by rendering with Blender[51]. For our object models, we choose to use Instant-NGP without a decoder, instead directly encoding the colour values in the feature grid. Furthermore, we remove view dependence for better multi-view consistency. Finally, we use lower resolutions for the object feature grids as compared to the scene feature grids. The object feature grids contain resolutions up to $128^3$ and $64^3$ features for the car and hydrant, respectively. Since our adversarial objective does not have any smoothness constraint, we found it critical to use lower resolution grids and remove the positionally encoded feature decoders to avoid aliasing effects.

### C.2   Driving Policy.

We train our own policy on which the attack will be performed. Our policy is an end-to-end RGB image neural network policy and the architecture is taken from [52]. We make a slight addition to goal condition the policy by adding a goal input to the linear component and increasing the capacity of the linear layers. The policy is trained via imitation learning, specifically DAgger [53], [54].

Expert actions are given by a lane following controller tuned for the simulator that gets access to the ground truth state, unlike the policy. The expert queried from various states random distances from the center of the road to recover from. Furthermore, random noise augmentation is used on the images during training to make the policy more robust to noisy observations.

### C.3   CARLA

We fit the background scene model using a dataset of 1800 images and their corresponding camera poses, which provide a dense covering of the CARLA scene.

When transferring our attacks back to the deployment scene, opacity values are usually not available. In order to evaluate our attacks, we assume that objects are opaque ($\alpha = 1$), and thus our method of blending in Equation B.3 can be calculated using just the depth and color values. We observe from experiments on the CARLA simulator that this type of composition is sufficient for the evaluation in the deployment environment.

**Driving Policy.** For our driving policy the initial training dataset of images is collected from the intersection in CARLA. We further fine-tuned the policy with some additional data collected from our surrogate simulator to ensure that our policy is not trivially failing due to slight visual differences. We use a total dataset of 120000 images in CARLA and 60000 images in the surrogate simulator in order to train the policy. We validated our policy on a hold out validation set consisting of 12000 images captured purely from the surrogate simulator. All data were collected by running the expert on the 3 reference trajectories. The policy was trained using behaviour cloning, where we gave examples of recovery from deviation by collecting data from random start locations around the nominal trajectory.

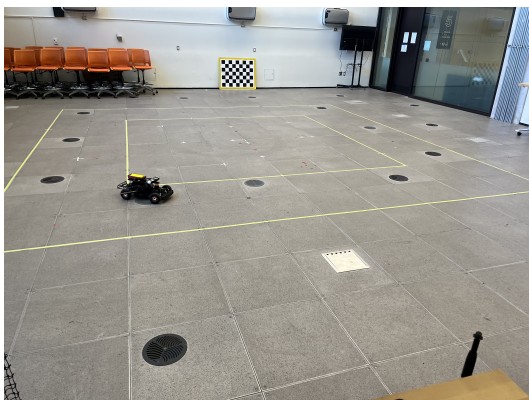

Figure C.1: Picture of driving area for the real world scenario experiments.

## C.4 Real World

We fit the background to a room in the real world using a dataset of 2161 images captured from an iPhone camera at 4K resolution. We collect data covering the room by walking around, then attach the iPhone to the robot to collect further data from the driving view points. The captured videos are processed using COLMAP [55, 56] for both camera intrinsic and the poses.

**Driving Policy.** We train a driving policy to track a square track in the room marked by green tape, this policy was trained using an expert PID controller with global positioning supplied by the VICON system providing $9584$ images. We further augment this again with $12000$ images from driving data in the NeRF scene. An overview of our working area is given in Figure C.1.

For all real world attacks we optimize the color of a cube in the surrogate NeRF scene, placed at one of the corners such that the camera will encounter this cube as the car takes the turn.

### C.4.1 Robot

We carry out experiments using the RACECAR/J[†] platform. The robot is equipped with a ZED stereo camera, of which we only utilize the RGB data from the left sensor, which has been configured to a resolution of 366x188 at 10 frames per second. We operate the robot inside a VICON system that positions the robot at a rate of 50Hz streaming through a remotely connected computer that runs policy as well as the image processing for some of the attacks.

### C.4.2 Green Screen Attack

For the green screen attack, we utilized a VICON system to accurately position both our robot and the green screen target. Using the green screen target position, as well as the camera parameters, we project one face of the cube on the input image to the policy. We opt to overlay the cube in such a manner to keep the policy driving in real time and to ensure that there is no penalty on control frequency. The image compositions is done at the remote computer where the controls are computed, which are then sent wirelessly to the robot to execute.

### C.4.3 Monitor Attack

To replace the green screen with a physical object, we place a monitor and display the same attack as above on the monitor. We place the monitor in a location such that it is visually consistent with the NeRF and green screen attacks. For the monitor attack, we utilize a 27-inch monitor with a 16:9 aspect ratio. Since the adversarial objects optimized in earlier examples are cubes we only use the center of the monitor to display the attack.

---

[†]https://racecarj.com/

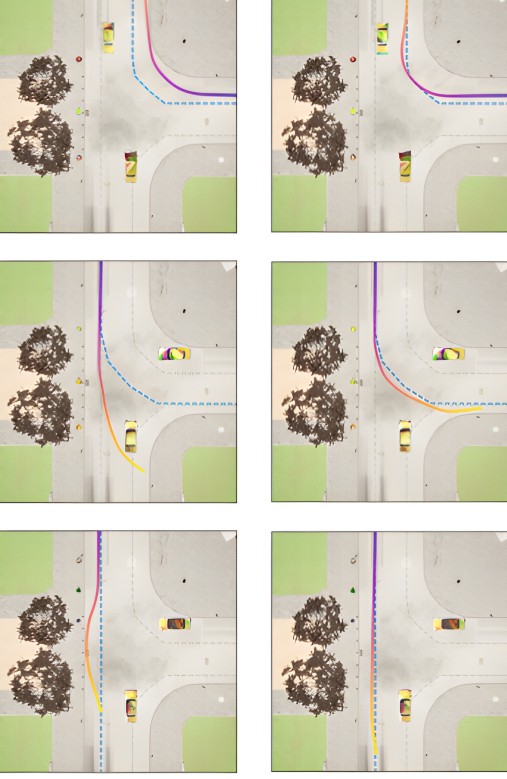

Figure D.1: The performance of the driving policy before (left) and after (right) retraining on the discovered adversarial scenarios.

| Scenario | CARLA Unperturbed | Attack Transfer in CARLA Random | Gradient | CARLA Attack After Retraining Gradient |
|---|---|---|---|---|
| Straight | 1166 | $1193 \pm 19$ | $1702. \pm 160$ | 1250 |
| Right | 1315 | $1476 \pm 12$ | $2101. \pm 75$ | 1307 |
| Left | 1448 | $1158 \pm 163$ | $2240. \pm 574$ | 1419 |

Table 2: Comparison of the total cross-track error for the retraining experiment over the 3 different trajectories. Results are extending the results from the main paper Table1 shown for the following cases: (1) no attack in CARLA (unperturbed), (2) an attack in the CARLA scene, (3) an attack in the CARLA scene after the driving policy is retrained using adversarial data.

# D    Appendix: Additional Experimental Results

## D.1    Incorporating Discovered Adversarial Scenarios in the Training Set

Our primary focus in this paper was to discover adversarial attacks for the evaluation of pretrained self-driving policy. Here we perform some preliminary investigations on fine-tuning our self-driving policies, on the old data and the adversarial attacks we found. Specifically, we take the attacks discovered by the gradient-based optimization and use them to collect additional imitation learning data. The collection is performed in the CARLA simulator using the depth compositing approach to insert the adversarial objects, as was done for the evaluation in the main paper. Apart from the object compositing, the data is collected in the same way as the original CARLA data used to train the base policy. We collect 24000 total frames over three trajectories with two different starting points. After fine-tuning our policy on the combination of the original dataset and the new adversarially augmented dataset, we evaluate the fine-tuned agent in the same scenario. We visualize the trajectories of the fine-tuned policy in Figure D.1 and report on the total deviation compared to before fine-tuning in Table 2. We find that the policy is no longer susceptible to the adversarial attacks, even though the initial starting position for evaluation was unseen during training.

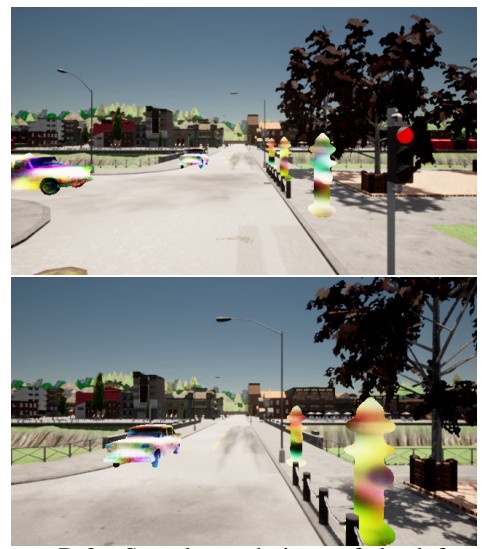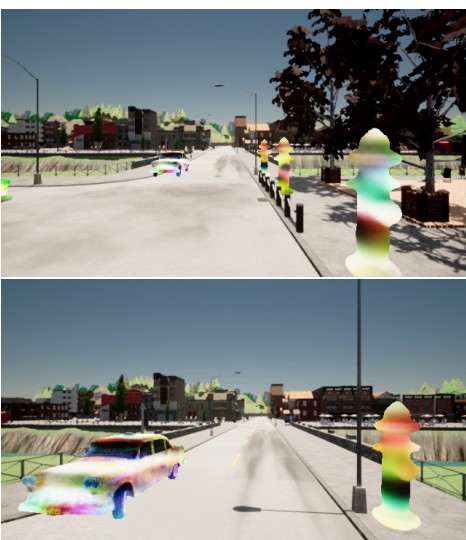

Figure D.2: Sample renderings of the left turn trajectory with the adversarial perturbations in CARLA from the ego vehicle's point of view. Four different snapshots from the evolution of the trajectory are shown.

## D.2 CARLA Visualizations

We show first person visualizations of our discoverered adversarial attacks inserted back into the CARLA deployment simulator in Figure D.2. We note the smoothness of the texture discovered by our method. Purely perceptual single-frame attacks typically exhibit a much higher frequency texture.

We show additional overhead trajectory views of adversarially attacked trajectories from one CARLA scene in Figure D.3.

## D.3 Object Translation Attacks

We find in practice that the loss landscape with respect to the poses of inserted objects are extremely non-smooth, as seen in Fig. D.4 We therefore investigated a mixed attack in NeRF where we use multi-start (multi-seed) gradient optimization for the object poses and single-seed gradient optimization for the adversarial textures. For this experiment, we used a more robust policy trained with the data from our color attacks, as well as additional examples of the adversarial objects in random poses. We report the results in Table 3. During the course of optimization we randomly sample a set of 5 poses within the constraints, then we perform 10 gradient descent iterations to refine the candidate solutions. We keep the best solution across 50 total evaluations. We see that even with a more robust policy, by combining gradients and multi-seed sampling we are able to discover some significant and transferable adversarial scenarios that incorporate both textures and object poses. We note that the straight trajectory is particular robust after the retraining, which causes the new gradient attacks to be less effective. We observe that modifying the poses of adversarial objects in addition to their textures allows the attacks to transfer even better to the deployment CARLA scene.

| | CARLA | NeRF | | Attack Transfer in CARLA | |
|---|---|---|---|---|---|
| Scenario | Unperturbed | Multistart Gradient | Random Only | Multistart Gradient | Random Only |
| Straight | 1248 | $1377 \pm 87$ | $1265 \pm 21$ | $1194 \pm 10$ | $1164 \pm 23$ |
| Right | 1648 | $2682 \pm 175$ | $2529 \pm 169$ | $2707. \pm 342$ | $1981 \pm 170$ |
| Left | 1353 | $1523 \pm 27$ | $1476 \pm 27$ | $1808 \pm 401$ | $1792 \pm 447$ |

Table 3: Total cross track error for attacking CARLA policy with random and gradient combined.

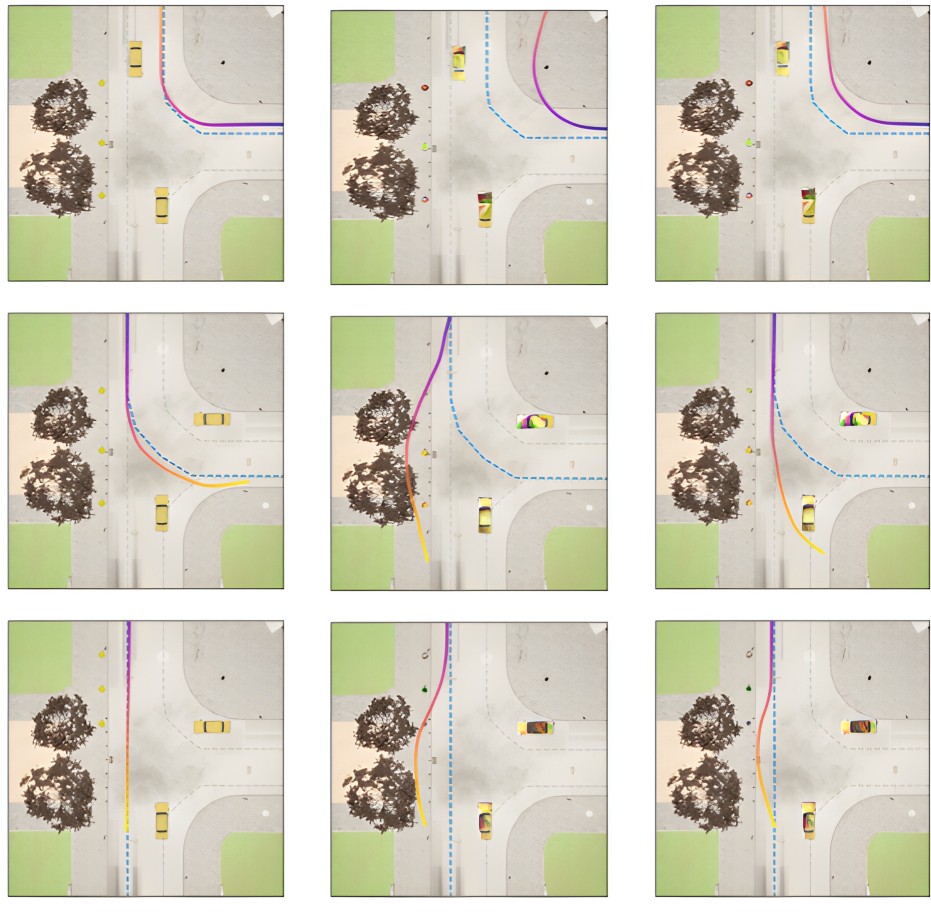

(a) Unperturbed        (b) Attacks in NERF        (c) Transferred

Figure D.3: Overhead views of three distinct trajectories driven by the policy. (a) shows the policy driving behavior in CARLA when no adversarial perturbation is introduced. (b) shows the policy driving behavior in the surrogate simulator with the discovered adversarial perturbation. (c) shows the same perturbation transferred to the deployment scene.

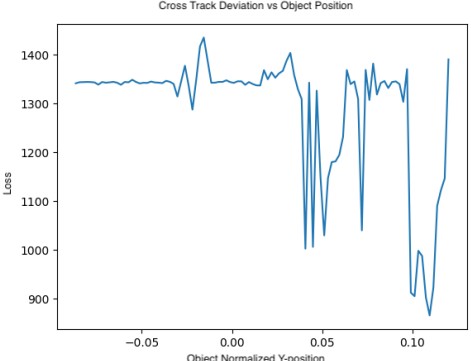

Figure D.4: Loss landscape as a function of a car position for the straight scenario

| CARLA | |
|---|---|
| Scenario | Differentiable Rasterization |
| Straight | $1647 \pm 38$ |
| Right | $1727 \pm 41$ |
| Left | $3397 \pm 180$ |

Table 4: Results of optimizing the texture of an adversarial mesh directly using PyTorch3D differentiable rasterization

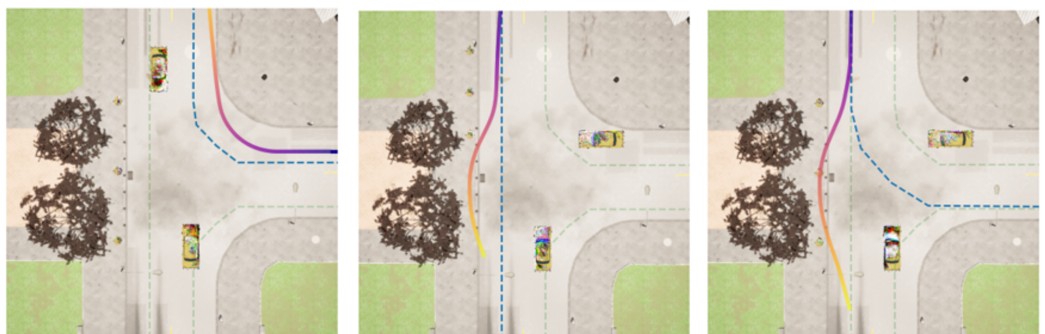

Figure D.5: Overhead views of the car driving with adversarial PyTorch3D objects.

### D.4  Differentiable Rasterization for Adversarial Textures

In this section, we perform an investigation on our method using different differentiable rendering methods for the inserted objects. Instead of carrying out volume rendering, we use PyTorch3D[57] to carry out differentiable rasterization for our adversarial objects. We optimize the textures of car and hydrant models as in our experiments on Instant-NGP, however, we now render the objects using PyTorch3d directly on top of CARLA background. We report the mean and standard deviation from 5 random seeds in table 4. Figure D.5 shows selected overhead views of the adversarial trajectories. We observe that using differentiable rasterization we are able to obtain results similar to volume rendering. However, there still needs to be work done to determine how to use differentiable rasterization to generate adversarial attacks that are transferable to the real world.

### D.5  Real-world Visualizations

We show aligned visualizations of the same adversarial real-world monitor attack in Figure D.6.

## E  Limitations

### E.1  Non-differentiable policies

In our work, we indeed assume that the visual driving policy is end-to-end differentiable. Recent work has shown the potential of modular end-to-end learned policies that can optimize the interaction between perception and planning [32, 34].

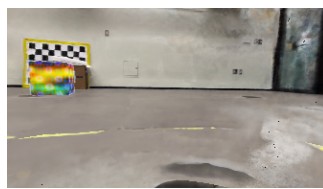 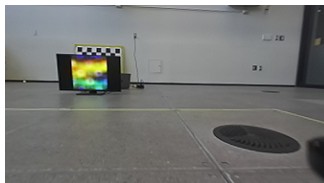 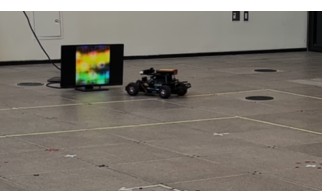

(a) Surrogate Simulator      (b) First person view      (c) Third person view

Figure D.6: Real-world adversarial monitor attack visualizations.

However, our current method does have a clear limitation when it comes to non-differentiable policies. Here we outline a few approaches to address this issue. If the policy is entirely black-box, then just as we have learned a differentiable surrogate scene to approximate the true simulator, it may be possible to also learn a differentiable surrogate policy to approximate the behavior of the non-differentiable policy [58]. Otherwise, we may need to resort to zeroth order optimization. This would prove computationally challenging for high-dimensional parameter spaces, but methods such as sparse Bayesian optimization could be applicable [59]. We can also leverage gradient information for Bayesian Optimization [60], or recent work on combined zeroth-order and first-order gradient estimators, such as [61] or [62].

If we consider the structure of the policy, most modern driving pipelines will still contain large chunks of end-to-end differentiable components, such as perception modules, together with non-differentiable components. In these cases, the policy would be a mixed computation graph where some parts are differentiable and others are not. Gradient estimation in mixed stochastic computation [63] graphs has been explored by many works in the contexts of probabilistic programming, variational inference and RL, and could be adapted for stochastic optimization methods. Gray-box bayesian optimization [64] also considers splitting up functions into constituent parts, and techniques for leveraging gradient information for BO exist as well.

Although parts of the driving policy may not be trivially differentiable via standard backpropagation, there are also many techniques (such as implicit differentiation and informed perturbations) proposed for obtaining gradients of algorithms including combinatorial [65] and convex optimization [66] that may be present in the self-driving policy. These could account for many of the "non-differentiable" components of modern driving policies, and it is worth exploring the applications of such techniques to them.

