# OpenReview forum: "Generating Transferable Adversarial Simulation Scenarios for Self-Driving via Neural Rendering"
_robot-learning.org/CoRL/2023/Conference — CoRL 2023 Poster_

### Official Review · Reviewer_Hzbf · 2023-07-17

**Confidence:** 4
**Originality:** Good
**Technical Quality:** Very Good
**Clarity Of Presentation:** Very Good
**Impact:** 3

**Recommendation:**

Weak Accept: I recommend accepting the paper, but will not argue for my recommendation if the majority of other reviewers have a different opinion.

**Review:**

This paper is well written and easy to follow - the figures and equations are clearly laid out and allow the reader to understand quickly their pipeline.

While none of the individual components are technically "novel" in isolation, the combination of recent neural surrogate techniques, adjoint gradient computation, and adversarial simulation are combined in a way that I think makes a sensible contribution to assessing image-based AV safety.

The related literature does a decent job placing the work within the context of adversarial sensor attacks, and general adversarial robustness. However, I wonder if the scope is a little too narrow? Ultimately this work is about safety and risk, in the sense that it is concerned with how an upstream image/sensor perturbation can affect downstream trajectory control. For this reason, I would have liked to have seen how the authors envision the work as fitting in to the wider story of perceptual-control safety and testing. For example, with respect to [Perception Error Models](https://ieeexplore.ieee.org/abstract/document/10161501), [Adaptive Stress Testing](https://ieeexplore.ieee.org/abstract/document/9981724), [Cyber-Physical System Falsification](https://arxiv.org/abs/1703.00978) etc.

The authors differentiate their work from other forms of adversarial attacks by the fact that their perturbations are:

- Temporally consistent
- Transferable
- Object Centric

This is commendable. However, I was left wondering if all this is really a sufficient description for "Realistic" attacks? Ultimately, from the images provided, it seems like the attacks that their neural renderer produces take the form of rainbow-colored plasma-like billboards. How do the authors see their work in the context of papers which attempt to make AV-perception systems more robust to types of corruption that are [actually likely to manifest in deployment](https://users.ece.cmu.edu/~koopman/pubs/pezzementi18_perception_robustness_testing.pdf)?

Similarly in the experiments - I think the work here is cleanly laid out and does a good job demonstrating the merits of their approach. However, in a broader context, is "maximizing cross-track error" the same as "making the trajectory unsafe"? The author's somewhat allude to this as the "true" goal when they talk about the significance of perturbations and whether the attack "causes to vehicle to under-steer and exit the track". I wonder if the work could benefit from relating it to work on more [Risk-Aware](https://papers.nips.cc/paper_files/paper/2022/hash/40739b3bb584c117b3e2f418d17f63a1-Abstract-Conference.html) notions of attack?

**Quality Of The Limitations Section:**

Limitations are addressed clearly

**Questions For Rebuttal:**

Listed above in the review. The paper is promising, and my main questions generally ask the authors to discuss how they see they work as fitting in to the broader landscape of safety-testing and assurance within the Automated-Driving domain (rather than solely "Neural Network Robustness").

**Robotics Focus:**

Sufficient demonstration on hardware

**Summary Of Paper:**

This paper takes recent work on differentiable neural rendering, and applies it to the task of generating adversarial scenarios for self-driving cars.

It does so by taking an differentiable image-based control policy, feeding it image observations from a neural rendering model and generating a trajectory via a simulator. The realized trajectory is then compared to a reference trajectory, and an optimization step attempts to adversarial change the parameters of the neural renderer such that the cross-track error between the reference and realized trajectories is maximized.

Their experiments (On both simulated CARLA deployments and a real-world RC track) show that their system is able to consistently find stronger adversarial examples than random-search (in terms of maximizing cross-track error), though attack strength diminishes when transferring results to a real-world monitor.

**Summary Of Recommendation:**

This paper shows a sensible application of NeRF technologies to a downstream AV-robustness task. While each of the individual components of their pipeline are not necessarily novel in isolation, and while the adversarial images produced are not "realistic" in the sense of what might be seen on a real highway, I think this is a decent first step in showing how such neural renderers have promise in terms of improving robustness for image-based control.

---

### Official Review · Reviewer_8f1g · 2023-07-17

**Confidence:** 4
**Originality:** Excellent
**Technical Quality:** Excellent
**Clarity Of Presentation:** Excellent
**Impact:** 4

**Recommendation:**

Strong Accept: I recommend accepting the paper and will argue for my recommendation even if other reviewers hold a different opinion.

**Review:**

I enjoy reading this pioneering work. It is well-motivated and easy to understand from the teaser to the experiments. In addition, the method is well-explained in Fig.3, which clearly describes how the method works. The experiments provide strong evidence that it can work on real robots.

The only suggestion I have is to add some details and move some content in the appendix into the main paper. For example:
1. Please mention that the system dynamic is differentiable by using Dubin's vehicle in the caption of Figure.3.
2. It would be better to include Appendix C3.1 in the main paper to show an important downstream application.





**Quality Of The Limitations Section:**

Limitations are addressed clearly

**Questions For Rebuttal:**

N/A

**Robotics Focus:**

Sufficient demonstration on hardware

**Summary Of Paper:**

End2end driving policies, like black-box, lack interpretability, and thus methods for discovering the failure cases are important for improving their safety. This paper proposes a promising method to attack a well-trained policy and efficiently discover object textures that confuse the policy. Concretely, the gradient can be found to maximize the error between the generated trajectory and the reference trajectory. After that, this gradient is fed into a differentiable renderer to modify the scene representation to the extent that policy won't work when taking this new adversarial representation as input. The experiments on both the simulator and the real robot show the effectiveness of the method.

**Summary Of Recommendation:**

A potentially impactful work. Recommend for acceptance.

---

### Official Review · Reviewer_qikM · 2023-07-20

**Confidence:** 3
**Originality:** Good
**Technical Quality:** Good
**Clarity Of Presentation:** Good
**Impact:** 4

**Recommendation:**

Weak Accept: I recommend accepting the paper, but will not argue for my recommendation if the majority of other reviewers have a different opinion.

**Review:**

The paper addresses an important problem in autonomous driving, specifically adversarial scenario generation. The application of NeRF for adversarial rendering is an interesting concept that will be widely applied in the future. It is interesting to see that authors even tried to reproduce (to the best of their abilities) generated adversarial examples on the real scene. Overall, the presence of robotics experiments is very appreciated.
Nonetheless, the paper has its limitations. I will mention them in the "Questions for Rebuttal" section.

**Quality Of The Limitations Section:**

Limitations are addressed clearly

**Questions For Rebuttal:**

- The major assumption of having differentiable dynamics, policy, and objectives may limit the applicability of the method to a narrow range of modern autonomous vehicle (AV) stacks.

- Using deviation from the desired trajectory as the sole objective may be misleading. For instance, if a human driver encounters a car flashing extreme colors, deviating from the initial desired trajectory to create a safe distance from the object might be the appropriate response. Under such a loss function, good behavior could become indistinguishable from poor one. This situation is complicated by the differentiability of the objective in my previous point, i.e. expressing more complex objectives could be a challenge.

- The scope of the work is somewhat limited, as acknowledged by the authors. It only focuses on manipulating object textures and poses. It would be beneficial to see the capability to manipulate the scene in a more sophisticated manner.


**Robotics Focus:**

Sufficient demonstration on hardware

**Summary Of Paper:**

The paper addresses the issue of adversarial scenario generation in the context of autonomous driving. The approach taken involves optimizing the textures and poses of objects in the scene to maximize the deviation of the ego vehicle from desired trajectories. Instead of developing a differentiable simulator, the work leverages NeRF as a neural renderer/simulator. The authors validate their method by testing it in CARLA and also on a toy car, demonstrating the transferability of the generated attacks to target domains.

**Summary Of Recommendation:**

The interesting direction of work and demonstration on a real robotic platform make me inclined toward accepting the paper.
Nonetheless, I admit that the paper shows a pretty limited scope and demonstration of its potential, hence I would not strongly object to rejection either.

---

### Official Review · Reviewer_i9eu · 2023-07-27

**Confidence:** 4
**Originality:** Very Good
**Technical Quality:** Very Good
**Clarity Of Presentation:** Very Good
**Impact:** 4

**Recommendation:**

Weak Accept: I recommend accepting the paper, but will not argue for my recommendation if the majority of other reviewers have a different opinion.

**Review:**

## Strengths
### Significance & Originality
I like the idea of using NeRF as a differentiable renderer for improving the robustness of self-driving policies. To the best of my knowledge, this is a novel application of NeRF. In the future, this method can potentially become very practical as more and more teams are using neural rendering to build photorealistic simulators.

### Figures
I like Figures 1-3. They illustrate the results, method, and system clearly.

### Quality
I think the proposed method is technically sound and claims are supported with experiments.

## Weaknesses
### Limitation
- Does the proposed method assume an end-to-end, differentiable, neural network policy? If so, I think this is a major limitation that is not mentioned in section 6. Lots of established methods for self-driving are not necessarily fully differentiable.

### Writing
I think the writing can be drastically improved.
- In section 3, line 101 - line 132 can be safely skipped and instead, just say we use instant-NGP with equation A.1 for rendering. Also, authors should consider moving equation A.1 to the main text as it is later referenced in line 188. This way, readers don’t need to go to the appendix to search for the equation.
- Lines 134-140 can be moved to section 4.3 when it is actually about to be used. Also, by deleting lines 101-132, authors can move more details of the implicit differentiation from the appendix to the main text so that readers can actually understand the mechanics behind the scene instead of just having rough ideas.


**Quality Of The Limitations Section:**

Additional details required

**Questions For Rebuttal:**

- Does the proposed method assume an end-to-end, differentiable, neural network policy?

**Robotics Focus:**

Sufficient demonstration on hardware

**Summary Of Paper:**

This paper proposes a pipeline to generate adversarial examples for end-to-end self-driving policy. Specifically, it first builds a neural radiance field of the streets as a surrogate, differentiable, photorealistic simulator. Then, it solves an optimal control problem in order to solve the perturbation that could lead the policy to deviate from its original route. In the experiments, the authors demonstrate that the proposed attack can be transferred to the real-world environment.


**Summary Of Recommendation:**

I think the originality of the paper is good and hence recommend a weak accept. I believe a rewriting of section 3 and clarification of the limitation will make the paper a good contribution to the community.

---

### Author Response · Authors · 2023-08-11
**Common Responses to Reviewer Questions**

We express our sincere thanks to all the reviewers for their generous time and effort in providing us valuable feedback. We believe the problem of automated and transferable adversarial scenario generation is very important and are glad that all the reviewers agree. We are also encouraged to see that all reviewers found the direction of NERF-based simulation to be promising in this area.

We will summarize our response to the main points here and will copy to each reviewer directly as well if space permits. We are happy to reply to any other questions or concerns during the discussion period as well.The reviewers raised great points and areas of related works. We will make an effort to include relevant discussions in the paper as well but due to the limited space, we may add these to the appendix.

### Does the policy need to be differentiable? (i9eu, qikM)

In our work, we indeed assume that the visual driving policy is differentiable. Recent work has shown the potential of modular end-to-end learned policies that could leverage synergies between vision and planning, such as [neural motion planning](https://arxiv.org/abs/2101.06679) and [others](https://arxiv.org/abs/2003.06404).

Nonetheless, we agree with the reviewers about the limitation. We are working on extending our results to be able to lift this assumption. We are currently considering a few possibilities:

1. Just as we have learned a differentiable surrogate scene to approximate the true rendering simulator, we can learn a differentiable surrogate policy to approximate the behavior of the non-differentiable policy.

2. We can use zeroth order optimization, which in general would be computationally challenging for high-dimensional parameter spaces but methods such as [sparse bayesian optimization(BO)](https://arxiv.org/abs/1301.1942) could be applicable. We can also [leverage gradient information for BO](https://arxiv.org/abs/1703.04389), or recent work on combined zeroth-order and first-order gradient estimators, such as [this](https://arxiv.org/abs/2202.00817) or [this](https://arxiv.org/abs/2112.13835).

3. If the policy contains a subset of components that are differentiable and a subset that is not, then we can use methods amenable for mixed computation graphs including [stochastic computation graphs](https://arxiv.org/abs/1506.05254) and [gray-box bayesian optimization](https://ieeexplore.ieee.org/document/9715343).

We updated our paper’s limitations section to explicitly state our requirement for a differentiable policy as of now, and included Appendix section D to describe in detail possibilities for how to address this issue.

### How to Discover Realistic Perturbations (qikM, Hzbf)

First, we added appendix section C.2 to the paper that performs object translation optimization, making the attacks more useful in practice. One challenge for using gradients in this front [(also identified in other literature)](https://arxiv.org/abs/2207.00167) is the rough loss landscapes that can arise from rendering and long horizon simulation.

Second, we agree that realistic texture attacks are important to handle. Despite being 3D and temporally consistent, we currently do not constrain our attacks to look realistic. We believe it is still meaningful to ensure AV systems are robust to such “non-realistic” adversarial cases when these attacks are able to transfer to the real world, but agree about the importance of ensuring realism. One way to take into account the realism of the scenario is by optimizing in the [latent space of a generative model of traffic behavior.](https://arxiv.org/abs/2112.05077). The resulting objective is still differentiable and amenable to gradient optimization, so we can use it in our loss function as a regularizer without any major changes. We can also leverage recent NERF literature that [models distributions of objects](https://arxiv.org/abs/2109.01750) or [lighting conditions](https://arxiv.org/abs/2008.02268).

### Cross-Track Error and Safety (qikM, Hzbf)

We understand that large cross-track error doesn’t always coincide with an unsafe scenario. When the policy’s primary objective is to do lane following on roads that other vehicles might be using, however, large cross-track errors can lead to unsafe situations by deviating from the intended behavior. There are other metrics that can model different aspects of unsafe behaviors, which we can use in our method without any conceptual changes: for example, probability of collision, time to collision, distance to the closest object, all of which we can approximate using distance to collision in a differentiable manner, and as such we can optimize them using our method. There are also types of unsafe behaviors that are generally hard to characterize in a loss function, for example, ego agent behaviors that cause other agents to enter into collision even if the ego agent does not. We think this is a very important area of research complementary to our contributions.

---

### Decision · Program_Chairs · 2023-08-30

**Decision:**

Accept (Poster)

**Comment:**

The paper presents a novel and promising approach to generating adversarial examples for end-to-end self-driving policy. The method first builds a neural radiance field (NeRF) of the streets as a surrogate, differentiable, photorealistic simulator. Then, it solves an optimal control problem to find the perturbation that can lead the policy to deviate from its original route. The method is evaluated on both simulated and real-world environments, the results demonstrate that the proposed attack can be transferred to the real-world environment.

Most reviewers agree that the paper is well-written and presents a novel approach to generating adversarial examples for self-driving policies. Specifically the strengths pointed out include
- The use of NeRF as a differentiable renderer for improving the robustness of self-driving policies.
- The figures and equations are clearly laid out and allow the reader to understand quickly their pipeline.

The reviewers also brought some weakness including
- The assumption of having differentiable dynamics, policy, and objectives may limit the applicability of the method to a narrow range of modern autonomous vehicle stacks.
- Using deviation from the desired trajectory as the sole objective may be misleading.
- The scope of the work is somewhat limited, as acknowledged by the authors. It only focuses on manipulating object textures and poses.

During the rebuttal phase, the reviewers  acknowledged the authors did a thorough job with clarifying and improving the paper in line with reviewer concerns.

For the above reasons, we recommend accepting this paper as poster. The authors should address the limitations of their method, such as the assumption of differentiable dynamics, policy, and objectives in the camera-ready version.